# A 16<sup>th</sup> century *Escherichia coli* draft genome associated with an opportunistic bile infection

George S. Long [1,2,13✉], Jennifer Klunk[1,2,3,13], Ana T. Duggan[2,4], Madeline Tapson[2,3], Valentina Giuffra[5], Lavinia Gazzè[6], Antonio Fornaciari[7], Sebastian Duchene [8], Gino Fornaciari[7], Olivier Clermont[9], Erick Denamur [9,10✉], G. Brian Golding[1] & Hendrik Poinar [2,11,12✉]

*Escherichia coli* – one of the most characterized bacteria and a major public health concern – remains invisible across the temporal landscape. Here, we present the meticulous reconstruction of the first ancient *E. coli* genome from a 16<sup>th</sup> century gallstone from an Italian mummy with chronic cholecystitis. We isolated ancient DNA and reconstructed the ancient *E. coli* genome. It consisted of one chromosome of 4446 genes and two putative plasmids with 52 genes. The *E. coli* strain belonged to the phylogroup A and an exceptionally rare sequence type 4995. The type VI secretion system component genes appears to be horizontally acquired from *Klebsiella aerogenes*, however we could not identify any pathovar specific genes nor any acquired antibiotic resistances. A sepsis mouse assay showed that a closely related contemporary *E. coli* strain was avirulent. Our reconstruction of this ancient *E. coli* helps paint a more complete picture of the burden of opportunistic infections of the past.

[1] Department of Biology, McMaster University, Hamilton, Canada. [2] McMaster Ancient DNA Centre, Departments of Anthropology and Biochemistry, McMaster University, Hamilton, Canada. [3] Daicel Arbor Biosciences, 5840 Interface Drive, Suite 101, Ann Arbor, MI 48103, USA. [4] Department of Anthropology, McMaster University, Hamilton, ON, Canada. [5] Division of Paleopathology, Department of Translational Research and New Technologies in Medicine and Surgery, University of Pisa, Via Roma 57, 56126 Pisa, Italy. [6] Department of Human Science (DISUM), University of Catania, Piazza Dante 32, 95124 Catania, Italy. [7] Department of Civilisations and Forms of Knowledge, University of Pisa, Via Trieste 40, 56126 Pisa, Italy. [8] Department of Microbiology and Immunology, Peter Doherty Institute for Infection and Immunity, University of Melbourne, Melbourne, VIC, Australia. [9] Université Paris Cité, IAME, UMR 1137, INSERM, 75018 Paris, France. [10] Laboratoire de Génétique Moléculaire, Hôpital Bichat, APHP, 75018 Paris, France. [11] Michael G. DeGroote Institute for Infectious Disease Research and CIFAR Humans and the Microbiome Program, Toronto, ON, Canada. [12] CIFAR Humans and the Microbiome Program, Toronto, ON, Canada. [13]These authors contributed equally: George S. Long, Jennifer Klunk. ✉email: longg2@mcmaster.ca; erick.denamur@inserm.fr; poinarh@mcmaster.ca

The recovery of ancient pathogen DNA (aDNA) from human victims has almost exclusively focused on historically significant mortality events, such as the Black Death, revealing the evolutionary history of canonical pathogens such as *Yersinia pestis*, *Mycobacterium tuberculosis*[1], and *Variola virus*. In contrast, much human morbidity and mortality is the result of opportunistic infections, ones that often remain invisible in the past. Opportunistic pathogens, those without historical records—such as *Escherichia coli*, *Pseudomonas aeruginosa*, and *Staphylococcus aureus*—have been understudied relative to their contemporary burdens on people today[2]. They are defined by their ability to infect during periods of stress, imbalance, or disturbance while being otherwise commensal or saprophytic[3]. Opportunistic infections likely played an important role in human mortality in our shared past and thus have had a broader impact on human health than can be or has been measured.

A confounding factor in the identification of historically understudied pathogens is that the resultant infections were likely primarily opportunistic. That is, they colonized their host environments asymptomatically, leaving no identifiable pathologies and as such are not easily identified in human remains[4]. Ancient DNA studies typically focus on obligatory pathogens such as *M. leprae*, *Salmonella enterica*, and *Y. pestis* that are easily correlated with pathologically distinct, or historically relevant mortality events and as such are easily distinguished as foreign in ancient human metagenomic DNA read data[5–7]. Opportunistic infections have the added burden of authentication due to their ubiquity as environmental contaminants and of modern commensal strains. Historical identification of these pathogens would allow for a careful assessment of their evolutionary history and the commensal-pathogen continuum defined by gene content gain and loss as strains modulate toward or away from host virulence.

*E. coli* is one such pathogen. It is a common commensal bacteria found in vertebrate gut microbiome[8] that can also become an opportunistic pathogen under specific conditions[9]. *E. coli* has such a large impact on our health care systems that it is the subject of several vaccine development efforts to mitigate the effects of the most pathogenic strains[10]. Having been responsible for several food poisoning outbreaks and becoming a leading pathogen for deaths caused by antimicrobial resistance, *E. coli* is thus a key bacterium used in public health surveillance[11].

Global sampling of *E. coli* strains produces a tree with several unique phylogenetic groups and interspersed pathovars[12]. While the phylogenetic relationships between strains are constructed based on genetic similarity, pathovars are defined by the virulence traits of their members. In many cases, members of the same pathovar do cluster together in the same clade, however, as virulence genes can be acquired horizontally, some pathovars—like enteroaggregative *E. coli*—are distributed across multiple phylogenetic groups[12]. This striking diversity and the evolutionary transitory states among *E. coli* strains highlight their genomic plasticity and evolutionary versatility along this aforementioned continuum. Ancient *E. coli* genomes would provide useful insights into the forces that influence the emergence of commensalism and pathogenicity in bacteria. Here, we describe the reconstruction of a 16th-century *E. coli* genome characterized from the gallstone of an Italian noble—Giovani d'Avalos (1538–1586)—highlighting a genome with commensal characteristics.

## Results

### The mummified remains
In 1983, the mummified remains of several Italian nobles were recovered from the Abbey of Saint Domenico Maggiore in Naples, Italy (Supplemental Fig. 1). A careful paleopathological and histological survey of one of the individuals—Giovani d'Avalos (NASD1), a Neapolitan noble who died in 1586 at the age of 48 (further information can be found in the Supplementary Notes section of the Supplemental Materials)—revealed thickened gallbladder walls, Rokitanski-Aschoff sinuses, and several intact gallstones[13]. These features suggested that NASD1 may have suffered from chronic cholecystitis[14]. While not the only cause of cholecystitis, chronic bacterial infections from *E. coli*, *Bacteroides spp.*, and *Clostridium spp.* can lead to the formation of gallstones[15]. These infections are typically indicated by a brown pigment, as can be seen in the gallstone from NASD1[15] (Fig. 1a, b).

**Authenticating the ancient DNA**. Metagenomic profiles generated from the DNA extracts using `Kraken 2`[16] provided preliminary evidence for the substantial and increasing presence of *Enterobacteriales* in digest rounds 2 (outer layer), 3–4 (middle layer), and 5–6 (inner layer) from the gallstone while indicating its absence in the tissue samples of NASD1 (Supplemental Fig. 2). *Enterobacteriales* reads are also present in digest round 1 however they are comparable with those in the reagent blanks (Supplemental Fig. 2), thus likely to be a mixture of endogenous and contaminant DNA such as *Bradyrhizobium*. Species-level indication suggested that *E. coli* made up the largest proportion of reads identified as *Enterobacteriales*. Fortunately, *E. coli*-specific reads are virtually absent (≤0.002%) from the blanks and present in the second through sixth digests. In comparison, other taxa such as *Alcaligenes* and *Bradyrhizobium* were present in both the experimental samples and the blanks and are likely to be contaminants. Human assigned reads were present in all digests but were also detected in the blanks. *Klebsiella aerogenes* was also detected, however only in digest rounds 3–4. To assess the authenticity of the Human, *E. coli*, and *K. aerogenes* reads we examined the deamination and depurination kinetics (Fig. 1c, d and Supplemental Fig. 3).

**Reconstructing the ancient *E. coli* genome and putative plasmids**. The reconstructed ancient *E. coli* genome had a mean read depth of $28.46[28.31, 28.62] \times$, mean heterozygosity of $7.66 \times 10^{-4}[6.66 \times 10^{-4}, 8.66 \times 10^{-4}]$, 4446 genes when mapped to the *E. coli* pan-genome dataset and a total estimated size of 4050, 429bp when the detected genes are concatenated. Significant differences (two-sided *t*-test, $P = 7.951 \times 10^{-05}$) were detected when comparing read depths of the core (3122 genes, $28.73[28.61, 28.86] \times$) and accessory (1324 genes, $27.82[27.39, 28.26] \times$) genomes (Fig. 2a and Table 1); the same was true for their mean heterozygosity with the core at $2.74 \times 10^{-4}$ $[2.34 \times 10^{-4}, 3.15 \times 10^{-4}]$ and accessory genome with $1.92 \times 10^{-3}$ $[1.61 \times 10^{-3}, 2.24 \times 10^{-3}]$ (Supplemental Fig. 4). Eighty-three genes were identified as having a substantially (at least two standard deviations) deeper gene coverage. These results were then compared to the closely related A phylogroup *E. coli* genomes of FSIS11816402, an sequence type (ST) 4995 strain, and K-12 MG1655, an ST10 strain (Fig. 2b and Supplemental Fig. 5). Both genome comparisons returned a lower mean depth with a larger variance of coverage. This is especially true for FSIS11816402 as 25.24% of its contigs had a mean coverage of less than $1 \times$ (Supplemental Fig. 6).

We identified two potential plasmids within the assembled scaffolds with `MOB-suite`[17]: *E. coli* MDR_56 plasmid unnamed 5 (`CP019906.1`) and *S. flexneri* 1a strain 0228 plasmids (`CP012732.1`). Subsequent mappings of the *E. coli* reads back to these reference plasmids confirmed their presence in the ancient strain and authenticated their origin (Fig. 2c and Supplemental Figs. 5, 7). One plasmid, `CP019906.1` had a read coverage similar to that of K-12 ($20.95[20.80, 21.10] \times$) and 34 genes, whereas, `CP012732.1` had lower coverage

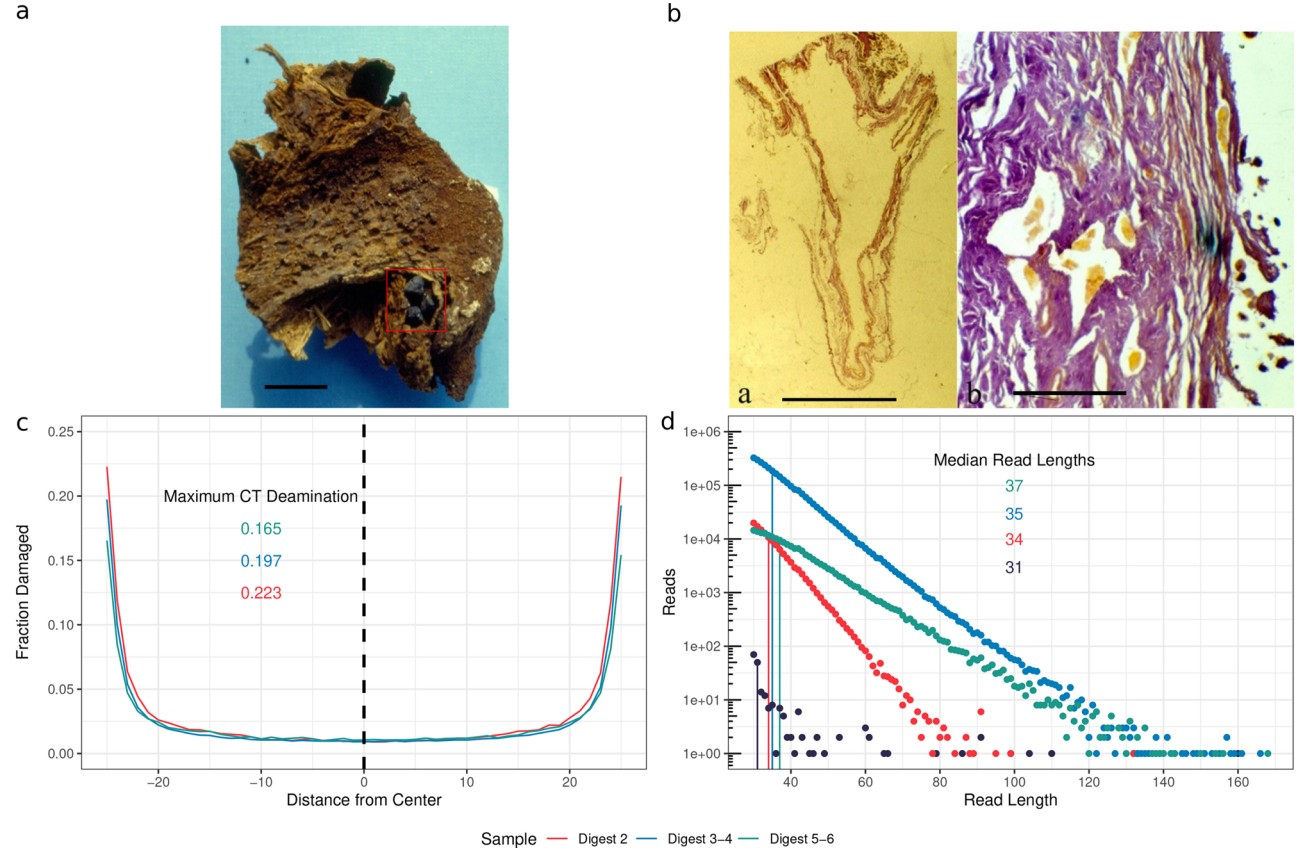

**Fig. 1 Characterization of both the physical sample and the authentication of the sequenced data. a** Liver and gallbladder of Giovani d'Avalos. The gallstones can be seen in the red rectangle. Note its dark brown coloring. Scale bar represents 1 cm **b** Gallbladder with a thickened wall (**a**) and Rokitansky-Aschoff sinuses (**b**) (Hematoxylin-eosin, 3X and 250X). The scale bar for (**a**) represents 2 cm and 100 μm for (**b**). **c** Damage plots of the 5′ and 3′ ends of mapped reads for *E.coli*. Colors refer to the digest, not damage type. `Mapdamage 2.0`[52] was used to calculate the damage rates. **d** Fragment length distribution of deduplicated mapped reads from *E. coli*. A $\log_{10}$ scale is used to emphasize the differences between the digests. A minimum length of 30 bp was required for a read to be kept.

(16.46[16.28, 16.64] ×) and only 19 genes. These plasmids contain regions with no read coverage—including a ~6.6 Kbp region in `CP012732.1`—but we could not confirm if the genes were part of the chromosome.

A subset of the sequenced reads were also classified as belonging to *K. aerogenes* by `Kraken 2`[16]. To confirm or refute this result, unmapped reads were aligned against the *K. aerogenes* (`NC_015663.1`) reference genome. The overall read depth (0.38[0.37, 0.38] ×) and genome coverage was low (3.43%), suggesting that only a subset of genes from *K. aeorgenes* was present. However, a 38 Kbp section of the genome (Fig. 2d) had a read mapping depth similar to the ancient *E. coli* results (20.36[20.23, 20.50] ×).

**Phylogenetic reconstruction**. In order to place the ancient *E. coli* within the global phylogeny and help identify the strain, we produced a core SNP alignment of 451 *E. coli* and four *Shigella* genomes representing the current breadth of *E. coli* diversity. After removing redundancy, the resulting phylogeny contained 107 genomes. This pruned sample set was then used to create a new alignment of 5007 core SNPs which returned an overall topology resembling previously published results and placed the ancient strain into a strongly supported phylogeny within phylogroup A (Fig. 3a).

To more carefully refine the position of the ancient strain within group A, we generated a reduced ML tree using 94 genomes (Fig. 3b) consisting of 291 SNPs. The ancient genome

clustered closely with the ST4995 strains with some statistical support (bootstrap support of 65%), signifying that it is likely part of the same sequence type (Table 2). An even further refined ML tree was then generated using the 22 available ST4995 genomes from `Enterobase`[18] (Fig. 3c). The core SNP alignment contained 16,026 nucleotides, providing a much greater resolution than the global and A0 subgroup alignments. The ancient strain clearly clusters within the ST4995 genomes.

We scanned for temporal signals across all phylogenies using `TEMPest`[19] and found one when the phylogeny was restricted to the ST4995 genomes (Supplemental Fig. 8), a result confirmed a date-randomization test in LSD ($P = 0.003$). The ST4995 phylogeny (Fig. 3c) suggests an 11th century (1027[787, 1220]CE) tMCRA for ST4995 strains, nearly 500 years before the diversification of the modern strains. The estimated evolutionary rate across the phylogeny was $2.555 \times 10^{-6}[1.567 \times 10^{-6}, 3.992 \times 10^{-6}]$ subs/site/year, similar to previously published results[20].

Multilocus sequence and typing of our ancient *E. coli* genome confirmed its placement in the ST4995—a rare sequence type with only 22 strains in the `Enterobase`[18] database which encompasses 182,476 entries[18]. It exhibited an Onovel15:H? serotype, similar to its sister taxa (see the Supplementary Discussion section in the Supplementary Materials for full results). A different *fimH* variant – *fimH*86 – was present in our *E. coli* strain.

**Global gene content**. The majority (95%) of the *E. coli* strains contained 3190 core genes, 3122 of which were detected in our

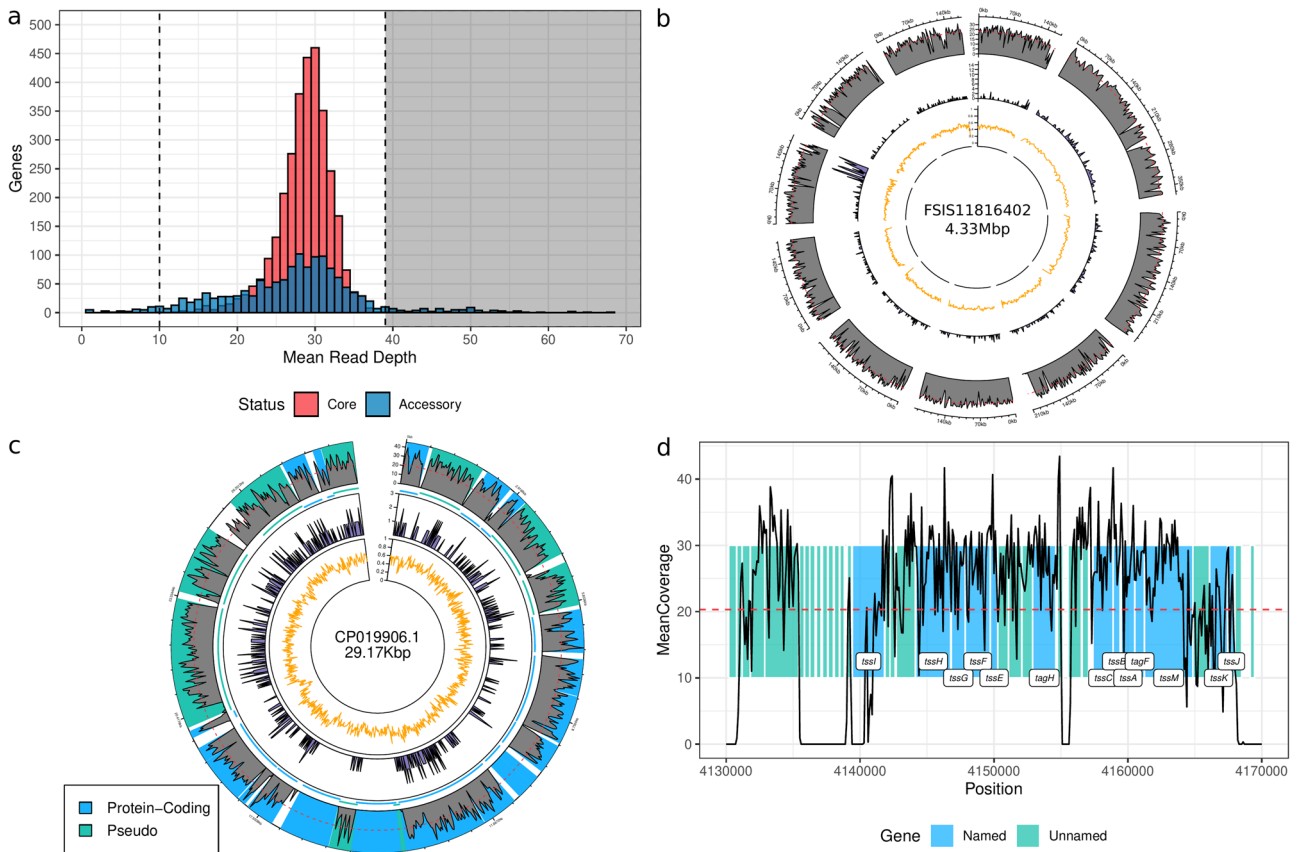

**Fig. 2 Read coverage metrics for the *E. coli* pan-genome in comparison to other genomes. a** Distribution of mean gene coverages with a CV ≤1 for the ancient genome. The dashed line indicates the detection threshold at 10×. The black rectangle on the right indicates the area where genes with a high copy number (as defined by $\bar{x} + 2 * s$). **b** Coverage plot for FSIS11816402. A window of 1% was used for illustration purposes. **c** Coverage plot for CP019906.1. A window of 0.1% was used for illustration purposes. The first track indicates the coverage with the red line illustrating the overall mean. The second track indicates the number of SNPs over the same window while the third is the GC content. **d** Gene coverage of the T6SS in *K. aerogenes* using a 100 bp window. Gene names are included when available.

**Table 1 Read coverage metrics for the *E. coli* pan-genome, FSIS11816402, K-12 MG1655, CP019906.1, and CP012732.1 references.**

| Sequence | % Positions with coverage ≥1× | Mean coverage |
|---|---|---|
| Pan-Genome | 97.93% | 28.46[28.31, 28.62] × |
| FSIS11816402 | 77.84% | 18.00[16.17, 19.83] × |
| K-12 MG1655 | 78.30% | 20.94[20.93, 20.95] × |
| CP019906.1 | 80.62% | 20.95[20.80, 21.10] × |
| CP012732.1 | 64.01% | 16.46[16.28, 16.64] × |

FSIS11816402 used the mean of means greater than 1× as it was an assembly. The 95% confidence interval is listed in the square brackets.

and previously published results[21]. The accessory PCoA indicates that the genome is a member of phylogroup A and an ancestral member of ST4995. The latter is confirmed by a P/A analysis of an ST4995-only pan-genome (Supplemental Fig. 10).

A total of 91 virulence factors were identified in the ancient genome, 37 of which were not found in *E. coli* K-12 MG1655 (see Table 3 for gene families with more than one hit). Type VI secretion system (T6SS) components consisted of the majority of these genes and contained high copy numbers. The T6SS—formed by the *tss* gene family as well as *hcp*, *vasK*, and *vgrG*—mediates antagonistic interactions between competing bacteria. In addition to its role in bacterial killing, T6SS is involved in interbacterial signaling, biofilm formation, and phage defense[22]. Parts of a type III secretion system were also detected at much lower levels.

We also observed incomplete virulence gene complements involved in several mechanisms. Several genes belonging to the *ecp* and *fim* fimbriae families were found in the ancient genome. While they are involved in virulence[9,23,24], they can also be present in commensal strains. Fimbriae known to be associated with virulence such as *cfaB* and *csg* were also identified[23] yet they do not provide enough information to pinpoint the specific pathovar of the ancient strain[9,23,24]. The key virulence factors for the *E. coli* pathovars such as *eae* and *stx* for Shiga toxin-producing *E. coli* (STEC), heat-labile and heat-stable genes for Enterotoxigenic *E. coli* (ETEC), and the type III secretion system in EPEC and EHEC were absent[9].

ancient genome (Supplemental Fig. 9). Enrichment analysis of the missing core genes revealed a significant lack of flagellar assembly genes ($P = 0.0445$), however, the protein-protein network contained the expected number of interactions ($P = 0.11$). An analysis of regions with mapped ancient reads in CP019906.1 revealed the presence of genes involved in biofilm formation (NP_311382.1) and a multidrug efflux pump (*acrD*). For CP012732.1, in contrast, reads mapped to pseudogenes are likely related to osmoprotection and transcription regulations.

A presence/absence (P/A) analysis of the accessory genome recapitulates the results from the ML phylogeny results (Fig. 4a)

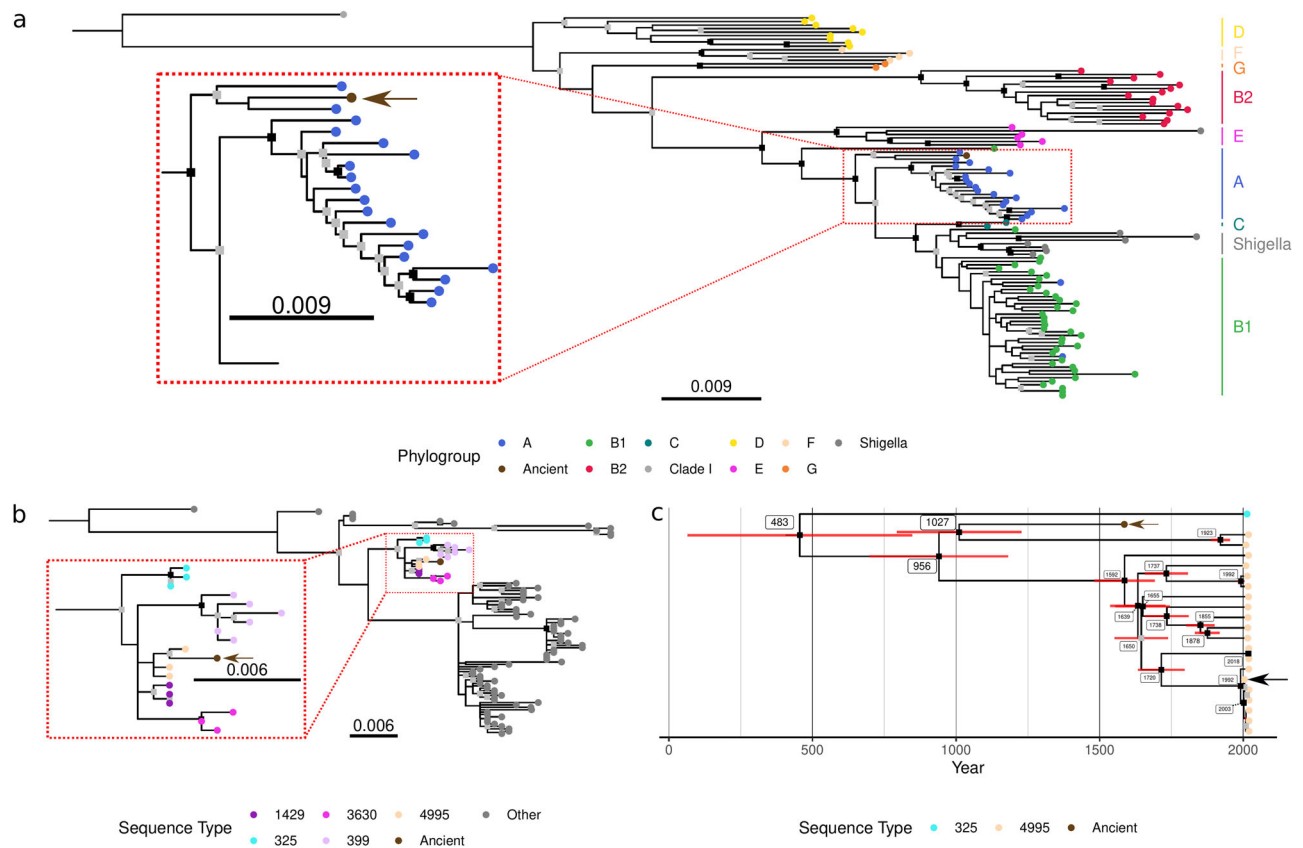

**Fig. 3 Maximum likelihood SNP phylogeny of *E. coli*. a** The global phylogeny with bootstrap values and phylogenetic groups with *E. coli* EC42405 as the out-group. **b** Phylogeny of the reduced subgroup A0 as defined by the Clermont genotype (+ - - -)[61]. Tip points represent the sequence type of the strain. IAI1 was the out-group of the phylogeny. **c** Phylogeny of the ST4995 strains. Red rectangles represent the 95% confidence interval for the topology; labels indicate the median date of divergence. The evolutionary rate for the phylogeny was $2.555 \times 10^{-6}[1.567 \times 10^{-6}, 3.992 \times 10^{-6}]$ subs/site/year. CFSAN051544 was the out-group. The brown arrows indicate the position of the ancient genome.

**Table 2 SNP distances of seven strains from the ST4995 phylogeny.**

| Genomes | CFSAN051544 | Ancient Ecoli | FSIS11816402 | ATCC 11229 | ESC_SA9272AA | ESC_DB2295AA | ESC_CA2237AA |
|---|---|---|---|---|---|---|---|
| CFSAN051544 | 0 | 771 | 869 | 920 | 971 | 1012 | 1024 |
| Ancient Ecoli | 771 | 0 | 419 | 503 | 554 | 596 | 607 |
| FSIS11816402 | 869 | 419 | 0 | 588 | 639 | 681 | 692 |
| ATCC 11229 | 920 | 503 | 588 | 0 | 215 | 219 | 230 |
| ESC_SA9272AA | 971 | 554 | 639 | 215 | 0 | 308 | 319 |
| ESC_DB2295AA | 1012 | 596 | 681 | 219 | 308 | 0 | 323 |
| ESC_CA2237AA | 1024 | 607 | 692 | 230 | 319 | 323 | 0 |

CFSAN051544 was the out-group of the phylogeny.

In contrast *astA*, a key virulence factor in Enteroaggregative *E. coli* (EAEC)[25], was detected in the ancient genome. In combination with evidence that T6SS is commonly found in EAEC strains[9], this may indicate that the ancient strain was been a member of this pathovar. The remaining EAEC virulence factors in the pan-genome—*aggA*, *aggR*, and *aap*—however, were not detected[25]. The ancient genome also contains a complete enterobactin system which is responsible for iron uptake[23]. The enterobactin system and *astA* gene were also found in *E. coli* K-12 MG1655, once again indicating that they do not confer enough virulence on their own.

To evaluate the extraintestinal virulence potential of ST4995 we tested a close relative (507 core SNPs separating the two strains) of the ancient genome, the reference strain ATCC11229 (see Fig. 3b and Supplemental Fig. 11 for similarities to the modern strain) in a well-calibrated mouse sepsis assay where ten mice were inoculated with the strain and their death recorded[26]. None of the mice were killed by ATCC11229, whereas all of the mice infected by the positive control B2 phylogroup CFT073 strain were killed. The phenotype of the strain which killed the mice is linked to the presence of specific extraintestinal virulence genes with a major role in the iron capture system encoding genes as those bared by the high-pathogenicity island (HPI)[27,28]. This result is in agreement with the absence of typical extraintestinal virulence genes present in the ATCC11229 and ancient strain.

We searched our ancient genome for the presence of antimicrobial resistance (AMR) genes using the `Resistance`

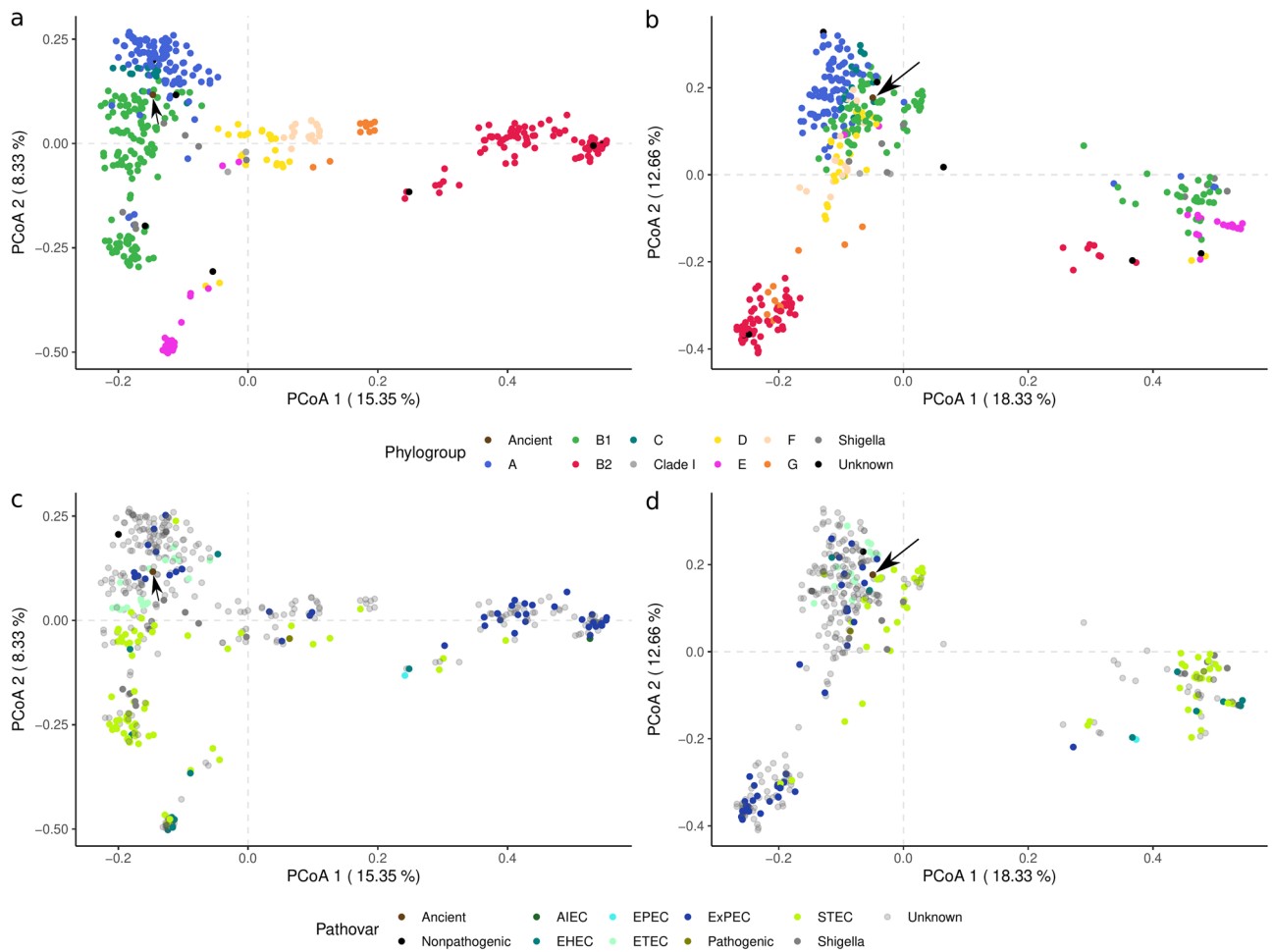

**Fig. 4 Principal coordinate analyses (PCoA) of gene presence/absence data.** The accessory genome (**a**, **c**) and identified virulence genes (**b**, **d**) were clustered using a binary distance. Phylogroups were identified using `Clermont Typing`[59] while pathovars were determined using available metadata from the `Patric`[49] database. The ancient genome is indicated by the black arrow. In both binary distances were calculated prior to create the PCoA.

**Table 3 Identified virulence factors which contained genes not found in K-12 MG1655.**

| Gene family | Genes | Mean coverage | Function | Pathovar |
|---|---|---|---|---|
| *tss* | 13 | 38.21[32.31, 44.10] × | Type VI Secretion System[9] | EAEC[9] |
| *elf* | 4 | 31.08[28.27, 33.89] × | Fimbriae[23] | EAEC, EHEC, ExPEC[23] |
| *hcp* | 3 | 36.02[22.49, 49.54] × | Type VI Secretion System[23] | EAEC, EHEC, ExPEC[9,23] |
| *cfa* | 2 | 23.01[5.01, 41.02] × | Fimbriae[23] | EAEC, EHEC, ExPEC[23] |
| *vas* | 2 | 51.62[49.80, 53.44] × | Type VI Secretion System[23] | EAEC[23] |
| *vgr* | 2 | 47.53[46.14, 48.91] × | Type VI Secretion System[23] | EAEC[23] |

*EAEC enteroaggregative E. coli, EHEC enterohemorrhagic E. coli, ExPEC extraintestinal pathogenic E.coli.*
*Only families which had more than one distinct gene are present.*

`Gene Identifier`[29] and found 47 distinct genes in the scaffold assembly of our ancient *E. coli* genome. Of these, 35 genes were also identified in the global pan-genome P/A analysis. Eight genes contained duplicates with *mdtB* being the most commonly detected. The remaining subset spanned thirteen antimicrobial drug resistance classes with nine classes represented more than once (Table 4). Resistance cassettes targeting the five most common drug classes were detected in the ancient *E. coli* strain. The majority of these genes are multidrug efflux pumps, which is typical of *E. coli*[30]. There were no unexpected AMR genes present in our ancient genome with all 35 being found in *E. coli* K-12 MG1655.

## Discussion

The DNA isolated from the stone showed clear evidence of DNA damage. Specifically, the deamination plots for the ancient reads mapping to the *E. coli* pan-genome reference contained characteristic aDNA profiles which indicated increased deamination rates at or near the terminal bases (Fig. 1c). In addition depurination kinetics plots of the *E. coli* mapped reads from libraries with sufficient *E. coli* read depth (digests 2, 3–4, 5–6) showed that fragments from internal sections of the gallstone (better protected from hydrolysis) were on average longer (34 bp median in digest 2 to 37 bp median in digests 5–6) (Fig. 1d). We obtained similar results for the *K. aerogenes* data across digests two through six.

**Table 4 AMR profile of the ancient _E. coli_.**

| Resistances | With efflux | No efflux |
|---|---|---|
| fluoroquinolone antibiotic | 17 | 1 |
| penam | 13 | 1 |
| tetracycline antibiotic | 10 | 1 |
| cephalosporin | 6 | 1 |
| peptide antibiotic | 5 | 3 |
| phenicol antibiotic | 5 | 1 |
| cephamycin | 4 | 1 |
| glycylcycline | 4 | 1 |
| rifamycin antibiotic | 4 | 1 |
| triclosan | 4 | 1 |
| carbapenem | 2 | 1 |
| penem | 2 | 1 |
| monobactam | 1 | 1 |

Antibiotic results from an RGI[29] search of the _E. coli_ scaffolds. A total of 29 genes were identified, with genes appearing multiple times. Only drug classes were detected at least once without the use of antibiotic effluxes. A single gene can be present in multiple drug classes.

Interestingly, the mapped human reads lack these signature features (Supplemental Fig. 3). More importantly, however, is that they confirm that the _E. coli_ and _K. aerogenes_ sequences are indeed ancient and were not the result of modern contamination.

Two potential explanations exist for the lack of deamination signal in the mapped human reads. The first is that the sample likely contains a mixture of both contaminating and to a lesser degree authentic ancient human DNA. This is almost certainly the case for the first digest, as it represents the outer layer of the gallstone, which would have been subjected to contamination from handling, storage, sampling, etc. However, as we moved to deeper layers of the gallstone, presumably better protected from contamination, the lack of endogenous human DNA was surprising. This is especially striking with the edit distances for both human and _E. coli_ reads from across the different digests (Supplemental Fig. 12). The more likely explanation is that there are simply not enough human DNA reads for meaningful damage analysis. Gallstones are typically formed from a combination of cholesterol, bile salts, and phosphatidylcholine[15]. Furthermore, in the highly acidic environment of the gallbladder, only bacteria that are specialized in surviving these conditions can thrive and potentially form a stone[15]. Since gallstones are not formed from direct human cellular components its interior is likely virtually devoid of human DNA, which is very different from the results we obtained from the DNA of ancient abscesses[31].

The gene depth results of the ancient _E. coli_ genome confirm the lack of modern contaminants as they are part of a unimodal normal distribution. The sole exception to these coverage stats are 83 genes with substantially larger mean depths (Supplemental Data 2). These genes consist of primarily hypothetical proteins (24.10%), integrases and transposons (19.28%), and the T6SS (13.25%). They also contain a significantly ($P = 5.03 \times 10^{-11}$) greater mean heterozygosity ($1.23 \times 10^{-2}[9.25 \times 10^{-3}, 1.53 \times 10^{-2}]$) than the rest of the ancient genome. These high copy number genes, however, are mainly accessory genes (96.34%) which suggests—in combination with the greater heterozygosity and presence of transposons—that they were located in mobile elements shared between multiple bacterial species.

The plasmids we detected in the scaffold assembly, CP019906.1 and CP012732.1, had lower gene depth coverages than that for our reconstructed chromosome (Table 1). Most plasmids are typically present in greater copy numbers than chromosomes[32], however, this was not the case as the ancient chromosome had a significant ($P = 0$ using Tukey's method to compare coverage estimates between the plasmids and ancient

chromosome) deeper mean coverage than CP019906.1 and CP012732.1. One possibility is that these two plasmids are actually not present, but rather the genes to which the ancient reads mapped were actually located in the ancient chromosome. This is supported by the fact that the mean gene depths for the plasmids fall firmly within the coverage distribution of the chromosome (Supplemental Table 2). This could explain large regions of the plasmids that lack any mapped reads, as well as the lack of reads mapping to intergenic regions (see Fig. 2c and Supplementary Fig. 5B).

The reads mapping to the _K. aeorgenes_ genome could represent a similar situation to the plasmid data. Interestingly, the 38 Kbp region in the _K. aerogenes_ chromosome which had a read depth similar yet significantly different ($P \approx 0$ using Tukey's method to compare coverage estimates between the _K. aerogenes_ T6SS and the ancient _E. coli_ chromosome) from the plasmids and _E. coli_ chromosome ($20.36 \pm 13.56 \times$) contains the Type VI Secretion System (T6SS), which is ubiquitous across Gram-negative bacteria and plays an important role in antagonistic interactions[22]. Given that T6SS is widespread among bacteria, its presence is not an exclusive marker for _K. aerogenes_. However, competitive read mapping between _E. coli_ and _K. aerogenes_ does suggest that this T6SS does indeed belong to _K. aerogenes_ rather than the homologous system in _E. coli_ suggesting a recent horizontal transfer, which is supported by a comparison of their relative GC contents[33] (Supplemental Fig. 13).

The structure of our global phylogeny resembles those previously published[34] and places the ancient genome in phylogroup A with meaningful support (100% bootstrap for a clade of three strains). Interestingly, other historical (~1800s) _E. coli_ strains have also been placed in this phylogroup[35]. Strains that have been isolated from humans living in less industrialized and more rural communities typically harbor strains falling within phylogroup A[8]. The late Medieval date of our sample, along with its phylogenetic placement helps confirm its authenticity. In addition, a study of _E. coli_ strains isolated from bile infections showed that they belong mainly to the phylogenetic A group[36].

We identified a temporal signal in the reduced phylogeny (Fig. 3c) which provides additional context for the evolutionary history of ST4995 strains. The majority of the sequence type can trace their lineage to a strain from approximately 1592[1479, 1690] CE, however, the divergence of our ancient genome and two others – ESC_AA9618AA_AS and ESC_VA4573AA_AS – has a deeper tMRCA ca. 956[689, 1172] CE. Given the disparity in ages between the two clades, it is likely that ST4995 is actually composed of at least two subgroups. Evidence supporting this can be seen in an MDS of the P/A analysis of all existing ST4995 genomes (Supplementary Fig. 9).

The P/A analysis of the accessory genes recapitulates the results from the ML phylogenies (Fig. 4a). The accessory PCoA confirms that the ancient strain is a member of phylogenetic group A while a subsequent P/A analysis with an ST4995-only pan-genome supports the proposition that the genome is a member of ST4995 (Supplemental Fig. 9). Phylogenetic group A—the group from which ST4995 stems—is a well-described clade and is the most commonly isolated type from human commensals[8]. _E. coli_ strain diversity is driven primarily by socioeconomic factors as individuals living in modern industrialized countries are more likely to carry B2 strains while those in less industrialized and rural communities harbor mostly A[8]. Given the time period NASD1 lived, our ancient _E. coli_ is a member of phylogenetic group A.

No acquired AMR genes were detected in the ancient genome confirming the age of the strain. Post removal of the multidrug efflux pumps from our genome, no other AMR genes were found when compared to _E. coli_ K-12 MG1655[30]. The latter is extremely pertinent as strong resistance to a particular drug class requires

several mechanisms[37]. Thus, it is likely that only mild resistances to these antimicrobial compounds were needed[30,38].

The identified ancient genome likely exhibited commensal characteristics as it shares most of its virulence factors with *E. coli* K-12 MG1655. The ancient strain lacks the canonical virulence factors for STEC and ETEC while also missing the components of the type III secretion system used in EHEC. Components of EAEC—*astA* and the T6SS[9]—were detected, yet other key virulence factors were missing and *astA* was also detected in K-12 MG1655. Results from the P/A analysis of the accessory genes with pathovar labels (Fig. 4c) and the virulence genes (Fig. 4d) present a similar picture. The ancient genome is positioned in a relatively general cluster containing a variety of different pathovars which include EHEC, ETEC, ExPEC, and STEC. The mouse sepsis model further reinforces that the ancient genome was an opportunistic pathogen as the proxy strain—ATCC 11229—did not infect any mice. This information, in combination with the identified virulence genes and P/A clustering of the ancient *E. coli* suggests that the ancient genome is an opportunistic pathogen that acquired a *K. aerogenes* T6SS gene cassette during expansion in the gallbladder, consistent with previous work showing *Klebsiella sp* often associated with *E. coli* in contemporary bile infections[39].

## Methods

**Sample, DNA extraction, library preparation, and sequencing**. We performed DNA extractions on a single gallstone—that had been isolated from the mummified remains of NASD1—in the clean-room facilities of the McMaster Ancient DNA Centre (McMaster University, Hamilton, Ontario, Canada) using methodologies specific to ancient DNA. We hypothesized that the successive mineralized layers of the stone would contain increasingly better-preserved DNA and thus subjected the gallstone to six successive rounds of demineralization and digestion[40]. Subsequently, DNA was extracted from 250 μL of the supernatant stemming from each round, which was then purified and prepared into dual-indexed libraries, size-selected to remove artifacts, and sequenced on the Illumina HiSeq 1500 platform with 90 bp paired-end reads[41,42]. Three additional tissue types from the same individual (bladder, small intestine, and lung) were processed and sequenced as above for comparative purposes. Full details of the methods can be found in the Supplementary Methods section in the Supplementary Materials.

**Bioinformatics**. Adapters were identified through `AdapterRemoval`[43] with trimming and merging performed by `leeHom`[44] using its aDNA setting. Sequencing lanes were pooled when applicable and `Kraken 2`[16] determined the overall metagenomic composition of the samples. This was done to characterize the metagenome and identify taxa of interest. A standard `Kraken 2`[16] database with some modifications was used—a *kmer* size of 25 bp and no minimizer—to account for the smaller read lengths of aDNA fragments. Reads underwent string deduplication using `prinseq`[45] prior to the metagenomic analysis.

Samples were then mapped against the human genome *GRCh38*[46] with `BWA`[47] with a maximum edit distance of 0.01 (`-n 0.01`), at most two gap openings (`-o 2`), and seeding effectively disabled (`-l 16500`)[48]. A minimum length of 30 bp and mapping quality of 30 was required for a read to be successfully matched. Fragments that did not successfully map were then compared with an *E. coli* pan-genome using the same settings. The pan-genome was created with *E. coli* genomes from `PATRIC`[49] and four *Shigella* genomes from NCBI. Specifically, genomes which were found in humans, chickens, cows, dogs, mice, or pigs and were determined to have a "good" genome quality (as defined by `PATRIC`) were selected resulting in 451 strains. These sequences were annotated with `Prokka`[50] using the proteins from *E. coli* K-12 MG1655 (NC_000913.3) obtained from NCBI as trusted sequences. The resulting annotations were compiled by `Roary`[51] to create the pan-genome. Paralogs were not split and a minimum `blastP` identity of 90% were used as additional settings. Unsuccessfully mapped human reads were also compared with *K. aerogenes* (NC_015663.1) as it was substantially— proportional abundance ≥1%—present in one of the digests while also being excluded from the blanks.

Successfully mapped human, *E. coli*, and *K. aerogenes* reads were deduplicated based on their 5′ and 3′ coordinates with `bam-rmdup` (https://bitbucket.org/ustenzel/biohazard-tools/src/master/). The deduplicated samples were pooled together based on their digest of origin (1, 2, 3–4, or 5–6). `MapDamage 2.0`[52] was used to estimate the amount of deamination across the mapped reads. Fragment length distributions (FLD) and mapping mismatches were also extracted. Heterozygosity was tested by calling single nucleotide polymorphisms (SNPs) using `bcftools` with ploidy set to two[53] and a quality threshold of 100.

Following authentication, an assembly was created with the complete set of mapped *E. coli* and *K. aerogenes* reads (i.e., not deduplicated) using `SPAdes`

3.14.1 in isolate mode[54] with customized *kmer* lengths—9, 19, and 29. The resulting assembly was an incomplete and highly fragmented scaffold. This was mostly due to the short median fragment lengths (35 bp) of the input reads and the required *kmer* sizes. Nevertheless, the N50 of the assembly was 939 bp with a total length of 3, 917, 339 bp which is substantially shorter than the complete *E. coli* chromosomes in `PATRIC` (5.00[4.96, 5.05]Mbp). `MOB-suite` was used to identify potential plasmids in the assembly, however, the final identification was done by mapping to the reference plasmids[17].

**Phylogenetic analysis**. We created an SNP alignment using the *E. coli* genomes from `PATRIC` and the *Shigella* genomes using `Snippy` (https://github.com/tseemann/snippy) with *E. coli* EC42405 (CP043414.1) as the reference strain. Non-canonical base pairs were replaced with an N and recombinant regions were removed with `gubbins`[55]. As the ancient genome contained several gaps the "filter_percentage" flag was increased to 32% to ensure that the assembly was not excluded. Once recombination sites were identified and removed a maximum likelihood phylogeny was created using `IQ-TREE 2`[56]. `ModelFinder` was used to select the appropriate phylogenetic model with ascertainment bias[57] and the tree was bootstrapped 1000 times. Redundant genomes were removed from the phylogeny using `Treemmer` making sure to retain 95% of the diversity—measured by Root-to-Tip length –, which returned 107 genomes[58]. A new SNP alignment and phylogeny was then created using these pruned genomes. The modern strains were labeled based on known *E. coli* phylogenetic groups using `ClermonTyping`[59]. Four *E. coli* strains which were identified as part of A were relabeled as potential *Shigella* strains. This is due to the presence of an *ipaH* gene in two of the genomes and their positioning in the phylogeny[60].

Upon determining the phylogroup of the ancient genome, the subgroup A0 *E. coli* genomes were used to carefully position the ancient strain within the same clade. This subgroup is defined by the Clermont genotype (+ - - -)[61]. An additional seven enterotoxigenic *E. coli* strains obtained from von Mentzer et al. 2014 were also included in the phylogeny[62]. This phylogeny was created using the same process noted previously and was rooted using IAI1 B1 (NC_011741.1). The phylogeny of the ST4995 strains was created with the 22 ST4995 genomes available in `Enterobase`[18] and rooted using the ST325 strain CFSAN051544 (SAMN05414627). To determine if the data contained any temporal signature, sampling dates or sequencing dates were assigned to the pruned phylogeny and a Root-to-tip (RTT) regression was performed for the global, A0 subgroup, and ST4995 phylogenies. The presence of a temporal signal was then assessed via a date-randomization test and with TEMPest[19]. A molecular clock was fit to the data using a least-squares dating approach using LSD[63,64].

The dated ST4995 phylogeny was created by dating the bootstrap trees with `LSD2` (using `-r l n-s 1500000` as settings)[63]. These trees were then summarized using `treeannotator`[65] with the height of the nodes set by the mean of the bootstrap distribution. The resulting phylogeny was then used to plot the dated phylogeny with the node height confidence intervals included.

**Gene function analysis**. To determine what genes were present in the ancient strain, we converted the read depths of the ancient *E. coli* pan-genome into a gene presence/absence (P/A) metric. We then used a conservative approach, identifying a gene as present if it had an average sequence depth of at least 10 × with a coefficient of variation (CV) ≤1. A CV filter was used to remove genes, which contained substantial regions of stacked reads. In comparison to a percent coverage threshold, a CV is more permissive with unmapped areas if the rest of the gene coverage is relatively consistent. An additional threshold of $\bar{x} + 2*s$ was used to identify genes with a high copy number. While still part of the ancient genome these genes could potentially belong to plasmids[32] or represent gene amplification[66].

The P/A matrix for modern strains was generated by `Roary`[51]. A 95% core genome of the modern strains was identified and genes missing from our ancient strain were submitted to `STRING`[67] to be functionally annotated using a network consisting of "high confidence" interactions. Virulence factors were identified using a curated set of known virulence genes[68]. Ambiguously named genes (i.e., `group_1000`) were run through `blastX`[69] against the RefSeq non-redundant protein database from 2019-11-29 with a maximum E-value of $10^{-5}$ to determine candidates. If multiple hits were found for a gene the smallest E-value was selected. Ties were resolved by choosing the gene with the highest bitscore followed by the highest percent identity. A minimum percent identity of 90% was required for a match to be kept and the ambiguous gene renamed. The descriptions of the previously ambiguous genes were searched for the keywords "`secretion`, `invasion`, and `enterotoxin`". Genes with hits from this search were included with the virulence genes. To determine the pathovar of the ancient genome, relevant information was extracted from the `PATRIC` metadata.

We performed a P/A analysis on the core, accessory genomes, and the virulence genes. The strains were transformed into a binary distance matrix and clustered using a PCoA. To ensure that redundant genes were not included, two exclusion criteria were created: genes that were ubiquitously present were excluded, and genes not present in at least five genomes were removed from the accessory genome analysis. Of the 451 *E. coli* genomes, four were removed from the pan-genome analysis due to low gene counts (see Supplemental Fig. 14 for gene counts and Supplemental Data 1 for removed genomes). The Resistance Gene Identifier

(RGI)[29] with default settings identified potential antimicrobial resistances in the ancient strain. Only RGI entries returned due to gene homologies and present in the pan-genome were accepted. Both the identified virulence and resistance genes were compared to the archetypal *E. coli* strain K-12 MG1655. It is avirulent and susceptible to antibiotics[70].

Ten 14–16 g (4-week-old) female mice OF1 from Charles River® (L'Arbresle, France) per strain received a subcutaneous injection of 0.2 ml of bacterial suspension in the neck ($2 \times 10^8$ colony-forming unit). Time to death was recorded during the following 7 days. Mice surviving more than 7 days were considered cured and sacrificed. The *E. coli* CFT073 strain was used as a positive control killing all the inoculated mice whereas the *E. coli* K-12 MG1655 strain was used as a negative control for which all the inoculated mice survive[26]. The protocol (n°APAFIS#4948) was approved by the French Ministry of Research and by the ethical committee for animal experiments. The ATCC11229 strain (AMC 198) was obtained from the Institut Pasteur collection (CIP 103795) and whole-genome sequenced using Illumina technology to verify the identity of the strain. This strain is of commensal origin and is commonly used in bacterial resistance testing.

**Reporting summary**. Further information on research design is available in the Nature Research Reporting Summary linked to this article.

## Data availability
The raw sequencing data has been uploaded to NCBI as PRJNA810725 with supporting metadata located in Supplementary Table 1. Access to the sample itself can be arranged by contacting G.S.L.

## Code availability
The scripts and additional data used to analyze the data after pan-genome mapping are available at https://github.com/longg2/AncientEcoli[71]. The scripts used for the initial mapping and pan-genome creation is available at https://github.com/longg2/LongBioinformatics/tree/vAncEcoli[72].

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

## Acknowledgements

We would like to thank Wael Elhenawy and Brian Coombes for their correspondence as well as Sara Dion for her technical assistance with the mouse model. Funding for this work was supported by an SSHRC Insight Grant, research funding from CIFAR's human and the microbiome program and the Boris Family Fund to H.P. as well as the NSERC grant RGPIN-2020-05733 awarded to G.B.G. This work was also partially supported by the "Fondation pour la Recherche Médicale" Equipe FRM 2016, grant number DEQ20161136698 to E.D.

## Author contributions

G.S.L. and J.K. were responsible for the investigation, methodology, formal analysis, and original draft of this study. G.S.L. also performed the data curation. M.T., A.T.D., V.G., L.G., and A.F. aided in the investigation. A.T.D. also contributed to the methodology. G.F. provided the sample. E.D. and O.C. validated the *E. coli* function and typing results. E.D. also provided funding for the mouse sepsis assay. S.D. validated the temporal signal and aided in its investigation. G.B.G. supervised and provided computational resources and funding. H.P. conceived and administered the project, provided funding, and also supervised. All authors approved the final manuscript.

## Competing interests

The authors declare no competing interests.
