## [Peer Review File · Communications Biology]

REVIEWERS' COMMENTS:

Reviewer #1 (Remarks to the Author):

The authors have provided a revised version of their article.
The clarity of the manuscript has been improved considerably.
All my comments were sufficiently addressed.
I have no additional comments.

A 16th Century *Escherichia coli* draft genome associated with an
opportunistic bile infection:
Second Round of Reviews

George S. Long^{1,2,*†}, Jennifer Klunk^{1,2,3,*}, Ana T. Duggan^{2,4}, Madeline Tapson^{2,3},
Valentina Giuffra⁵, Lavinia Gazzè⁶, Antonio Fornaciari⁷, Sebastian Duchene⁸,
Gino Fornaciari⁷, Olivier Clermont⁹, Erick Denamur^{9,10,†},
G. Brian Golding¹, Hendrik Poinar^{2,11,†}

April 4, 2022

Comments from the second round of review at Communications Biology. Responses are coloured red.

Reviewer 1

The authors have provided a revised version of their article.
The clarity of the manuscript has been improved considerably.
All my comments were sufficiently addressed.
I have no additional comments.

We'd like to thank for Reviewer 1 for taking the time to ensure that we addressed their previous concerns. We are glad that the message of the manuscript is clearer than it was during our initial submission.